# Investigating Therapeutic Effects of Indole Derivatives Targeting Inflammation and Oxidative Stress in Neurotoxin-Induced Cell and Mouse Models of Parkinson’s Disease

**DOI:** 10.3390/ijms24032642

**Published:** 2023-01-30

**Authors:** Ya-Jen Chiu, Chih-Hsin Lin, Chung-Yin Lin, Pei-Ning Yang, Yen-Shi Lo, Yu-Chieh Chen, Chiung-Mei Chen, Yih-Ru Wu, Ching-Fa Yao, Kuo-Hsuan Chang, Guey-Jen Lee-Chen

**Affiliations:** 1Department of Life Science, National Taiwan Normal University, Taipei 11677, Taiwan; 2Department of Neurology, Chang Gung Memorial Hospital, School of Medicine, Chang Gung University, Taoyuan 33305, Taiwan; 3Medical Imaging Research Center, Institute for Radiological Research, Chang Gung University/Chang Gung Memorial Hospital, Taoyuan 33302, Taiwan; 4Department of Chemistry, National Taiwan Normal University, Taipei 11677, Taiwan

**Keywords:** Parkinson’s disease, therapeutics, NLRP3 inflammasome, neuroinflammation, oxidative stress, MPP^+^ HMC3 cell model, MPTP mouse model

## Abstract

Neuroinflammation and oxidative stress have been emerging as important pathways contributing to Parkinson’s disease (PD) pathogenesis. In PD brains, the activated microglia release inflammatory factors such as interleukin (IL)-β, IL-6, tumor necrosis factor (TNF)-α, and nitric oxide (NO), which increase oxidative stress and mediate neurodegeneration. Using 1-methyl-4-phenylpyridinium (MPP^+^)-activated human microglial HMC3 cells and the sub-chronic 1-methyl-4-phenyl-1,2,3,6-tetrahydropyridine (MPTP)-induced mouse model of PD, we found the potential of indole derivative NC009-1 against neuroinflammation, oxidative stress, and neurodegeneration for PD. In vitro, NC009-1 alleviated MPP^+^-induced cytotoxicity, reduced NO, IL-1β, IL-6, and TNF-α production, and suppressed NLR family pyrin domain containing 3 (NLRP3) inflammasome activation in MPP^+^-activated HMC3 cells. In vivo, NC009-1 ameliorated motor deficits and non-motor depression, increased dopamine and dopamine transporter levels in the striatum, and reduced oxidative stress as well as microglia and astrocyte reactivity in the ventral midbrain of MPTP-treated mice. These protective effects were achieved by down-regulating NLRP3, CASP1, iNOS, IL-1β, IL-6, and TNF-α, and up-regulating SOD2, NRF2, and NQO1. These results strengthen the involvement of neuroinflammation and oxidative stress in PD pathogenic mechanism, and indicate NC009-1 as a potential drug candidate for PD treatment.

## 1. Introduction

Parkinson’s disease (PD), characterized by resting tremor, rigidity, bradykinesia, and postural instability, is the second most common neurodegenerative disorder affecting the elderly [1]. The pathological studies find a massive loss of dopaminergic (DAergic) neurons located in the pars compacta of the substantia nigra (SN) in the midbrain and subsequent depletion of dopamine in their projections [2]. The neurodegeneration of PD could be caused by a complex interaction of genetic and environmental factors [3].

Although the disease etiology remains to be clarified, it has been shown that oxidative stress contributes to neurodegeneration of PD [4]. A variety of environmental insults, including 1-methyl-4-phenyl-1,2,3,6-tetrahydropyridine (MPTP), specifically increase oxidative stress, damage DAergic neurons, and produce parkinsonism with symptoms like the main features of PD [5]. 1-Methyl-4-phenylpyridinium (MPP^+^), the toxic metabolite of MPTP, inhibits mitochondrial electron transport chain complex I (NADH:ubiquinone oxidoreductase) to compromise mitochondrial oxidative capacity [6]. In MPTP-treated mice, interleukin (IL)-1β, IL-6, and tumor necrosis factor (TNF)-α mRNA expression levels increase significantly both in the SN and caudate-putamen in comparison with untreated mice [7]. Lines of evidence also demonstrate that microglial activation, release of inflammatory factors, and IgG targeting of neurons may play important roles in the neurodegeneration of PD [8,9], and anti-inflammatory drugs provide a protective effect both in animal models and epidemiological studies [10]. Therefore, strategies or compounds that reduce oxidative stress and neuroinflammation may be beneficial to PD patients.

In the past years, our group has focused on screening novel synthetic compounds to test their therapeutic effects on several neurodegenerative disease models. Indole is an aromatic heterocyclic compound with a wide range of biological activities. Its chemical reactivity makes it a suitable candidate for modification, leading to the development of various novel derivatives with potential as drug candidates for the treatment of various diseases [11]. Previously, indole compound NC009-1 (C_19_H_16_N_2_O_3_) has been shown to have aggregation-reducing and neuroprotective effects by activating heat shock protein beta 1 (HSPB1) in tauopathy cell model [12] and spinal spinocerebellar ataxia (SCA) type 17 cell and mouse models [13]. In addition, NC009-1 activates apolipoprotein E (APOE) and neurotrophic receptor tyrosine kinase 1 (NTRK1) in amyloid beta (Aβ)-GFP SH-SY5Y cells, and Aβ precursor protein (APP)_Swe_/presenilin 1 (PS1)_M146V_/microtubule associated protein tau (Tau)_P301L_ triple transgenic Alzheimer’s disease (AD) mouse models [14], and decreases IL-1β-mediated pathway in SCA type 3 SH-SY5Y cell model inflamed with IFN-γ-primed HMC3 conditioned medium [15]. In the present study, we aimed to investigate the neuroprotective potential of NC009-1 and derivative compounds with methyl, phenyl or methyl formate substituent present on the benzene ring in MPP^+^-activated human microglial HMC3 cells and/or the sub-chronic MPTP-induced mouse model of PD.

## 2. Results

### 2.1. Indole Compounds and Cytotoxicity

Four synthetic indole derivatives NC009-1, -2, -3, and -11 were examined (Figure 1A). Based on molecular weight (MW: 320.12–396.15), hydrogen bond donors (HBD: 1), hydrogen bond acceptors (HBA: 5–7), and calculated octanol–water partition coefficient (cLogP: 3.60–5.40), NC009-1, -2, and -11 meet Lipinski’s criteria in predicting oral bioavailability (MW ≤ 450, HBD ≤ 5, HBA ≤ 10, cLogP ≤ 5) [16] (Figure 1B). In accordance with calculated polar surface area (PSA) of 63.60–107.21 Å^2^ (Figure 1B), all except for NC009-11 displayed potential for blood–brain barrier (BBB) penetration (<90 Å^2^) [17]. With BBB permeation score greater than that of threshold (0.02; [18]), all NC009 compounds were predicted to be BBB-permeable by an online BBB predictor (Figure 1B).

The cytotoxicity of NC009 compounds (1–100 µM) in human HMC3 microglial cells was examined by MTT assay after 24 h compound treatment. As shown in Figure 1C, all test compounds had cell viability greater than 80% in compound-treated HMC3 cells. These results demonstrated the low cytotoxicity of the test compounds.

The DPPH free radical scavenging activity and oxygen radical absorbance capacity of NC009 compounds were examined using antioxidant quercetin as a positive control [19]. While no DPPH radical scavenging activity was observed (Figure 1D), NC009 compounds at 20–100 μM concentrations displayed similar Trolox equivalent antioxidant capacity (NC009-1: 22–39 μM; NC009-2: 20–37 μM; NC009-3: 14–38 μM; NC009-11: 11–25 μM) based on the trapping of peroxyl radicals (Figure 1E). For comparison, all four NC009 compounds were included in the following HMC3 cell study.

### 2.2. Activation of HMC3 Microglia and Anti-Inflammatory Potentials of NC009 Compounds

MPP^+^, the active neurotoxic metabolite of the parkinsonian toxin MPTP, inhibits complex I of the mitochondrial respiratory chain, inducing energy depletion and producing reactive oxygen species [20]. MPP^+^ induces mouse microglial N9 cell activation [21]. To investigate the effect of MPP^+^ on human microglial cells, different concentrations of MPP^+^ (0–10 mM) were added to HMC3 cells for 20 h. As shown in Figure 2A, MPP^+^ at a concentration range of 0.15625–10 mM significantly decreased HMC3 cell viability compared to no MPP^+^ treatment (*p* = 0.028−<0.001). Treatment of 2.5–10 mM MPP^+^ significantly increased NO release in culture medium (*p* = 0.019−<0.001), suggesting the occurrence of inflammatory response in human microglial cells. Based on these results, 3 mM MPP^+^ was selected to test anti-inflammatory potentials of NC009 compounds. HMC3 cells were pre-treated with NC009 compounds (1–10 µM) for 8 h before MPP^+^ (3 mM) addition for 20 h (Figure 2B). As shown in Figure 2C, NC009-1 at 10 µM significantly increased cell viability (from 63% to 87%, *p* = 0.012). In addition, NC009-1 at 5–10 µM decreased release of NO in cell culture medium (from 4.1 µM to 2.8–1.7 µM, *p* < 0.001). The observation of reduced expression of CD68 (from 131% to 111%, *p* = 0.010) and CD11B (from 135% to 107%, *p* = 0.002), markers of activated microglia [22], also displayed the anti-inflammation action of NC009-1 in MPP^+^-activated HMC3 cells (Figure 2D). The anti-inflammatory potential of NC009-1 at 10 µM were further supported by the reduced release of IL-1β (from 82 pg/mL to 35 pg/mL, *p* < 0.001), IL-6 (from 1020 pg/mL to 759 pg/mL, *p* = 0.003), and TNF-α (from 59 pg/mL to 25 pg/mL, *p* = 0.004) in cell culture medium (Figure 2E).

It has been reported that MPP^+^ induced NLRP3 inflammasome activation in BV-2 microglia [23]. The protein levels of NLRP3, CASP1, iNOS, IL-1β, IL-6, and TNF-α with and without compound treatment in MPP^+^-stimulated HMC3 cells were also examined using a Western blot. As shown in Figure 2F, MPP^+^ addition increased the expressions of NLRP3 (164%; *p* = 0.040), CASP1 (141%; *p* = 0.173), iNOS (146%; *p* = 0.014), IL-1β (227%; *p* < 0.001), IL-6 (276%; *p* < 0.001), and TNF-α (202%; *p* = 0.008), whereas treatment with NC009-1 at a 10 μM concentration significantly reduced the expression levels of these markers involved in inflammasome and neuroinflammation pathways (NLRP3: 89%, CASP1: 66%, iNOS: 98%, IL-1β: 128%, IL-6: 125%, TNF-α: 84%; *p* = 0.012–<0.001).

### 2.3. Effect of NC009-1 on MPTP-Induced Motor Behavior in Mice

MPTP, a prodrug to the neurotoxin MPP^+^ which selectively destroys DAergic neurons in the brains, was frequently used to establish mouse model for PD [24]. Given the anti-inflammatory potential of NC009-1 in MPP^+^-activated HMC3 cells, we established a sub-chronic MPTP mouse model (Figure 3A) to examine the neuroprotective effects of NC009-1 for PD. On the gait test (Figure 3B), MPTP injection led to a shorter stride length at day 49 compared to the normal control (right front paw: 6.2 ± 0.5 vs. 7.2 ± 0.5 cm, *p* = 0.004; right hind paw: 6.0 ± 0.6 vs. 6.9 ± 0.5 cm, *p* = 0.034; left front paw: 6.0 ± 0.6 vs. 7.0 ± 0.7 cm, *p* = 0.023; left hind paw: 6.0 ± 0.8 vs. 6.9 ± 0.7 cm, *p* = 0.052). Treatment with NC009-1 markedly prevented the decrease of the stride length in right front paw (7.0 ± 0.7 cm, *p* = 0.017). For base width, MPTP injection led to a significant increase for both front paws (1.4 ± 0.0 vs. 1.3 ± 0.0 cm at day 35, *p* = 0.016; 1.4 ± 0.1 vs. 1.2 ± 0.1 cm at day 42, *p* = 0.025; 1.4 ± 0.0 vs. 1.2 ± 0.1 cm at day 49, *p* = 0.005) and hind paws (2.5 ± 0.1 vs. 2.3 ± 0.1 cm at day 35, *p* = 0.004; 2.6 ± 0.1 vs. 2.4 ± 0.1 cm at day 42, *p* = 0.004; 2.7 ± 0.1 vs. 2.5 ± 0.1 cm at day 49, *p* = 0.001) compared to the normal control. Treatment with NC009-1 markedly decreased base width at day 49 for both front paw (1.3 ± 0.1 cm, *p* = 0.040) and hind paw (2.6 ± 0.0 cm, *p* = 0.039).

The tail suspension test measures behavioral despair [25]. During a 6 min period, normal control mice spent 3.7 ± 0.5 min in passive immobility (Figure 3C). Mice tested at 49 days post-MPTP-lesioning spent significantly more time immobile (4.7 ± 0.4 min, *p* = 0.017). However, treatment with NC009-1 markedly decreased immobility at day 49 (4.0 ± 0.4 min, *p* = 0.023).

### 2.4. Effect of NC009-1 on Dopamine, DAT, 4-HNE/TH, IBA1, and GFAP Levels in MPTP-Treated Mice

In mice, MPTP treatment promotes the formation of reactive free radicals and reduces the production of dopamine [24]. By examining the dopamine levels of striatum using HPLC, the administration of MPTP significantly reduced the dopamine level (0.79 ± 0.46 μg/g striatum, *p* = 0.004) compared with controls (1.63 ± 0.51 μg/g tissue), while treatment with NC009-1 successfully rescued the reduction of dopamine level of striatum caused by MPTP (1.45 ± 0.42 μg/g striatum, *p* = 0.015) (Figure 4A). In addition, MPTP administration significantly reduced DAT level (31%, *p* = 0.025), and treatment with NC009-1 successfully rescued the reduction in striatum (92% vs. 31%, *p* = 0.045) (Figure 4B). Furthermore, administration of MPTP significantly up-regulated the oxidative stress marker 4-HNE in TH^+^ neurons in the ventral midbrain (4-HNE/TH: from 8% to 13%, *p* = 0.047), while treatment with NC009-1 successfully rescued the up-regulation of 4-HNE in TH^+^ neurons (6%, *p* = 0.017) (Figure 4C). The 4-HNE/TH ratio demonstrated correlation with the base width of front paws (day 49: *r* = 0.548, *p* = 0.012), the stride length of right front paw (day 49: *r* = −0.492, *p* = 0.028), or time of immobility (*r* = 0.468, *p* = 0.038).

We then further investigated the potential effect of NC009-1 on anti-neuroinflammation by examining the expression levels of IBA1 (microglial activation marker) and GFAP (astrocyte activation marker) in the ventral midbrain of MPTP-treated mice. MPTP increased IBA1 fluorescent intensity (from 17% to 28%, *p* = 0.012), while treatment with NC009-1 reduced this abnormal microglial activation (18%, *p* = 0.017) (Figure 4D). GFAP fluorescent intensity was also up-regulated by MPTP treatment (from 26% to 41%, *p* = 0.031), while treatment with NC009-1 reduced fluorescent intensity of GFAP (28%, *p* = 0.042). Taken together, NC009-1 reduces the neuroinflammation induced by MPTP in mice.

### 2.5. Down-Regulation of Neuroinflammation and Up-Regulation of Cellular Redox Signaling by NC009-1 in MPTP-Treated Mice

Up-regulated expression of mature IL-1β has been reported in the SN of an acute MPTP-induced PD mouse model and in cerebrospinal fluid of PD patients [26]. Moreover, MPTP-driven NLRP3 up-regulation in microglia plays a central role in DAergic neurodegeneration [27]. We thus examined the expression of inflammatory cytokines and NLRP3. As shown in Figure 5A,B, IL-1β (162%, *p* < 0.001), IL-6 (174%, *p* = 0.048), TNF-α (210%, *p* < 0.001), NLRP3 (216%, *p* = 0.028), CASP1 (203%, *p* = 0.025), and iNOS (159%, *p* = 0.046) were up-regulated in MPTP-treated mice, whereas treatment with NC009-1 down-regulated IL-1β (126%, *p* = 0.048), IL-6 (103%, *p* = 0.015), TNF-α (123%, *p* = 0.008), NLRP3 (132%, *p* = 0.048), CASP1 (115%, *p* = 0.047), and iNOS (90%, *p* = 0.025). The expression levels of IL-1β (113%), IL-6 (82%), TNF-α (107%), NLRP3 (133%), CASP1 (94%), and iNOS (76%), in NC009-1-treated mice were comparable to the controls.

In addition, cellular redox signaling including SOD2, NRF2, and NQO1 by NC009-1 in MPTP-treated mice were examined (Figure 5C). SOD2 (38%, *p* = 0.003), NRF2 (45%, *p* = 0.024), and NQO1 (36%, *p* = 0.005) levels were reduced in MPTP-treated mice, whereas treatment with NC009-1 up-regulated SOD2 (74%, *p* = 0.028), NRF2 (85%, *p* = 0.046), and NQO1 (81%, *p* = 0.047). Again, the expression levels of SOD2 (86%), NRF2 (91%), and NQO1 (98%) were not significantly altered between mice with and without NC009-1 treatment.

## 3. Discussion

Microglia, the innate immune responders of the central nervous system, are key mediators of neuroinflammation in neurodegenerative diseases [28]. The pronounced activation of microglia in the brains of PD patients up-regulates inflammatory factors, and increases reactive oxygen species (ROS) and neuroinflammation [29,30]. Here we demonstrate that indole derivative NC009-1 alleviates MPP^+^-induced production of inflammatory mediators in HMC3 microglia. By down-regulating neuroinflammation and up-regulating cellular redox signaling, NC009-1 ameliorates behavioral impairments, increases dopamine and dopamine transporter levels in the striatum, and reduces oxidative stress as well as activation of microglia and astrocytes in the ventral midbrain of the sub-chronic MPTP-induced mouse model. The study results strengthen the roles of oxidative stress and neuroinflammation in PD pathogenesis, suggesting the potential of NC009-1 for the treatment of PD.

Activated microglial infiltrations are observed in the SN of postmortem PD brain [31]. Pronounced microglial activation is also observed in *LRRK2* p.G2019S transgenic rats and *PLA2G6* knockout mice [32,33], indicating neuroinflammation as a common pathogenesis in both sporadic and genetic-defined PD. Pathological α-synuclein aggregation in PD can induce microglial activation and dysfunction. The extracellular α-synuclein fibrils can bind to toll-like receptor 2 and 4 (TLR2 and TLR4), and thereby activate NLRP3 inflammasome and its downstream CASP1 and IL-1β pathway [34]. The pathological study reveals that NLRP3 is up-regulated and co-localized with microglia in the SN of PD patients [35]. Levels of IL-1β are also elevated in the striatum of PD patients [36,37]. Higher serum levels of IL-1β are observed in PD patients compared to control subjects [38]. Sustained expression of IL-1β in the striatum causes DAergic neuronal death and motor disabilities in rats [39]. Small molecule NLRP3 inhibitor MCC950 decreases inflammasome activation and effectively mitigates motor deficits, nigrostriatal DAergic degeneration, and accumulation of α-synuclein aggregates in 6-hydroxydopamine (6-OHDA)- and α-synuclein fibrils-treated mice [35], indicating NLRP3 as a sustained source of neuroinflammation driving progressive DAergic neuropathology. In this study, the results of NC009-1 to down-regulate NLRP3 and IL-1β in MPP^+^-treated HMC3 microglia and MPTP-treated mice highlight its potential to reduce PD neuroinflammation and neurodegeneration.

The TLR-mediated signal pathway ultimately activates NF-κB, then triggers NO, IL-6, and TNF-α production [40]. iNOS is the primary source of NO [41]. Elevated levels of iNOS in glial cells have been found in PD brains [42]. Similar to our findings, up-regulated iNOS expression is also observed in animal models of PD treated with MPTP [43], 6-OHDA [44], and α-synuclein oligomers [45]. NO reacts with superoxide to generate the toxic free radicals such as peroxynitrite (ONOO^−^) and hydroxide anion (OH^−^) [46]. Treatment with the iNOS inhibitor GW274150 in 6-OHDA-treated mice or knockout of the iNOS gene in MPTP-treated mice protects animals against loss of DAergic neurons [44,47]. Our results showed NC009-1 also reduced the production of NO in HMC3 microglia and down-regulated iNOS in both MPP^+^-treated HMC3 microglia and MPTP-treated mice, indicating its neuroprotective role for PD via mediating nitrosative stress.

IL-6 causes neuronal death in neurodegenerative diseases [48]. The levels of IL-6 are elevated in striatum, cerebrospinal fluid (CSF), and serum of PD patients [49,50,51,52,53,54,55,56,57,58]. TNF-α activates microglia, causing progressive loss of DAergic neurons in the SN [59,60,61]. TNF-α is up-regulated in SN of PD patients [62]. The TNF-α levels in CSF are elevated in parkinsonian patients [59]. Serum levels of TNF-α are also elevated in PD patients [38,51,53]. Similar to our study, MPTP treatment up-regulates TNF-α in the striatum of mice [62,63]. Thalidomide, an inhibitor of TNF-α synthesis, reduces MPTP-induced neuronal damages in mouse striatum [63]. Knockout of *TNF-α* attenuates MPTP toxicity in mouse striatum as well [64]. Our results showed that NC009-1 reduced IL-6 and TNF-α in MPP^+^-treated HMC3 microglia and MPTP-treated animals, implying its neuroprotective mechanisms by regulating these pro-inflammatory factors.

The activation of microglia up-regulated the activity of NADPH oxidase to increase the production of ROS [28], which results in lipid peroxidation, and subsequently, generation of cytotoxic 4-HNE [65,66]. Anti-oxidative responses up-regulate a number of anti-oxidative factors, such as SOD2, NRF2, and NQO1, to reduce these radicals [67]. Increased levels of SOD have been reported in the frontal and motor cortex of PD patients [68,69]. NRF2 inactivation is observed in MPTP- or 6-OHDA-treated SH-SY5Y cells or mice [70,71]. The expressions of NRF2 and NQO1 in DAergic neurons derived from iPSCs carrying a *PARKIN* mutation are consistently down-regulated [72]. The levels of NRF2 in peripheral leukocytes are elevated in PD patients [73]. Antioxidants may have neuroprotective effects in PD by enhancing activities of SOD or NRF2 pathways. For example, gypenosides mitigate the MPTP-induced reduction of SOD activities in mouse SN [74]. Dimethyl fumarate, a potent NRF2 enhancer in treating multiple sclerosis, demonstrates neuroprotection against MPTP- and α-synuclein-induced neurotoxicity in mice through activating NRF2 [75,76]. Indole derivative NC001-8 also protects DAergic neurons derived from SH-SY5Y or iPSCs carrying a *PARKIN* mutation against MPP^+^ and H_2_O_2_-induced neurotoxicity by up-regulating NRF2 and NQO1 [71]. By up-regulating autophagy and the NRF2 pathway, disaccharides including trehalose, lactulose, and melibiose demonstrate neuroprotective effects against α-synuclein-induced neurotoxicity [77]. Our study demonstrated the anti-oxidative property of NC009-1 to reduce neurodegeneration in PD.

## 4. Materials and Methods

### 4.1. Compounds and Cell Culture

Indole derivatives NC009-1, -2, -3, and -11 were synthesized by iodine-catalyzed C-alkylation of indoles as previously described [78]. The synthesized compounds were examined by nuclear magnetic resonance spectroscopy. The test compounds stayed soluble in cell culture medium at concentrations up to 100 μM. In addition, quercetin was obtained from Sigma-Aldrich Co. (St. Louis, MO, USA) as a control for measuring radical scavenging capacity. Human HMC3 microglial cells (ATCC CRL-3304) were routinely maintained in Dulbecco’s modified Eagle medium/Nutrient mixture F-12 (DMEM/F-12) supplemented with 10% fetal bovine serum (FBS) (Thermo Fisher Scientific, Waltham, MA, USA). Cells were cultured in an incubator at 37 °C (NuAire, Plymouth, MN, USA) with 95% relative humidity and 5% CO_2_.

### 4.2. Bioavailability and BBB Permeation Prediction

Molecular weight (MW), hydrogen bond donor (HBD), hydrogen bond acceptor (HBA), octanol–water partition coefficient (cLogP), and polar surface area (PSA) of test compounds were analyzed and calculated using ChemDraw Ultra 12.0 http://www.perkinelmer.com/tw/category/chemdraw (accessed on 5 October 2020). In addition, prediction of blood–brain barrier (BBB) permeation was carried out using online BBB predictor version 0.90 https://www.cbligand.org/BBB/ (accessed on 5 October 2020).

### 4.3. Cytotoxicity Assay

To evaluate compounds’ cytotoxicity, HMC3 cells (2 × 10^4^) were plated on 48-well dishes, grown for 20 h, and treated with the test compounds (1–100 μM). The next day, 20 μl of tetrazolium dye 3-(4,5-dimethylthiazol-2-yl)-2,5-diphenyltetrazolium bromide (MTT, 5 mg/mL) (Sigma-Aldrich) was added to the cells at 37 °C for 3 h. The resulting insoluble purple formazan products were dissolved by lysis buffer (10% Triton X-100, 0.1 N HCl, 18% isopropanol). The absorbance of the dissolved product at optical density (OD) 570 nm was read using a FLx800 fluorescence microplate spectrophotometer (Bio-Tek, Winooski, VT, USA).

### 4.4. Antioxidant Assay

1,1-Diphenyl-2-picrylhydrazyl (DPPH) (Sigma-Aldrich), a stable free radical for measuring the antioxidant activity of different samples, was used to measure the free radical-scavenging activity of the studied indole compounds. Briefly, DPPH solution (100 µM) was prepared in ethanol. After adding test compounds (10–160 µM), the mixture was vortexed for 15 s and allowed to stand for 30 min at room temperature. Subsequently, the mixture was measured spectrophotometrically at 517 nm (Multiskan GO microplate spectrophotometer; Thermo Fisher Scientific). The free radical scavenging activity was calculated as the percentage of DPPH discoloration using the formula 1—(absorbance of sample/absorbance of control) × 100%—with EC_50_ calculated using the interpolation method.

The test for oxygen radical absorbance capacity [79] was performed using OxiSelect™ kit (Cell Biolabs, San Diego, CA, USA). Briefly, serial dilutions of Trolox standard (2.5–50 μM) and test compounds (4–100 μM) were prepared in 50% acetone. After adding fluorescein to blank (50% acetone), standards or samples, the mixture was mixed and incubated at 37 °C for 30 min. Subsequently, free radical initiator 2,2′-azobis(2-methylpropionamidine) dihydrochloride (AAPH) was added to produce peroxyl radicals (ROO•). The quenching of fluorescent probe over time was recorded for 60 min, with excitation at 480 nm and emission at 520 nm wavelengths (Bio-Tek FLx800). To quantify the oxygen radical absorbance capacity in a sample, the area under the curve (AUC) for blank, standards, and samples were calculated. After subtraction of the blank, the equivalent Trolox concentrations of samples were expressed based on the Trolox calibration curve.

### 4.5. HMC3 Microglia Activation and Inflammatory Mediators Detection

HMC3 cells were plated into 6-well (2 × 10^5^/well) dishes, grown for 20 h, and treated with MPP^+^ (0–10 mM) (Cayman, Ann Arbor, MI, USA) for 20 h. The cell viability was evaluated by MTT assay as described, and the release of nitric oxide (NO) in cell culture medium was evaluated by Griess assay according to manufacturer’s protocol (Thermo Fisher Scientific). To examine anti-inflammatory potential of test compounds, HMC3 cells were pre-treated with these compounds (1–10 µM) for 8 h before MPP^+^ (3 mM) addition for 20 h, and cell viability and NO release were examined as described.

In addition, HMC3 cells with 10 µM compound/3 mM MPP^+^ treatment was fixed with 4% paraformaldehyde (PFA) for 30 min, permeabilized with 0.1% Triton X-100 for 10 min, and blocked with 2% bovine serum albumin (BSA) for 20 min. Immunocytochemistry (ICC) staining of HMC3 cells was performed using primary anti-cluster of differentiation 68 (CD68, 1:1000; Cell Signaling #76437, Danvers, MA, USA) or anti-integrin subunit α M (CD11B, 1:1000; eBioscience #14-0118-82, San Diego, CA, USA) antibody at 4 °C overnight, followed by donkey anti-rabbit Alexa Fluor 555 (1:1000; Thermo Fisher Scientific #A-31572) or goat anti-mouse IgG (H+L) Cy5 (1:1000; Zymed Laboratories #81-6516, South San Francisco, CA, USA) secondary antibody staining for 2 h at room temperature. Nuclei were detected using 4′,6-diamidino-2-phenylindole (DAPI; 0.1 μg/mL; Sigma-Aldrich). After staining, cells were imaged and analyzed using ImageXpress Micro Confocal System (Molecular Devices, Sunnyvale, CA, USA). All fluorescence image intensities were normalized to the number of DAPI-positive signal cells to compute the mean cell integrated intensity. Three independent experiments were performed in triplicate, with around 7500 cells analyzed from each sample.

The levels of IL-1β, IL-6, and TNF-α in medium pre-treated with 10 μM compound were determined using enzyme-linked immunosorbent assay (ELISA). Specifically, human Instant IL-1β, IL-6, and TNF-α ELISA^TM^ kits (Invitrogen, Carlsbad, CA, USA) were used, according to the experimental procedures supplied by the manufacturer. The OD at 450 nm was detected using Multiskan GO spectrophotometer (Thermo Fisher Scientific).

To examine NLRP3 inflammasome activation, HMC3 cells were lysed using buffer (50 mM Tris-HCl pH 8.0, 2 mM EDTA pH 8.0, 150 mM NaCl, 0.5% sodium deoxycholate, 0.1% SDS, 50 mM NaF, and 1% NP40) containing the protease inhibitor cocktail (Sigma-Aldrich). After sonication, the lysates were centrifuged (12,000× *g* for 10 min at 4 °C) and protein concentrations determined (Bradford protein assay; Bio-Rad, Hercules, CA, USA). Aliquots of protein (20 µg) were separated by electrophoresis using 10% SDS-polyacrylamide gel followed by transfer to a polyvinylidene fluoride (PVDF) membrane. After being blocked, the membrane was stained with NLR family pyrin domain containing 3 (NLRP3) (1:500; Cell Signaling #15101s), caspase-1 (CASP1) (1:500; Cell Signaling #3866s), inducible nitric oxide synthase (iNOS) (1:500; Cell Signaling #20609), IL-1β (1:200; Cell Signaling #12242s), IL-6 (1:500; Cell Signaling #12153s), TNF-α (1:500; Abcam #ab9739, Cambridge, UK), or glyceraldehyde-3-phosphate dehydrogenase (GAPDH) (1:1000, MDBio #30000002, Taipei, Taiwan) primary antibody at room temperature 2 h or 4 °C overnight. The immune complexes were detected using horseradish peroxidase (HRP)-conjugated goat anti-mouse (#GTX213111-01) or goat anti-rabbit (#GTX213110-01) IgG antibody (1:5000; GeneTex, Irvine, CA, USA) and chemiluminescent HRP substrate (Millipore, Billerica, MA, USA).

### 4.6. Sub-Chronic MPTP Mouse Model

The animal experiments were conducted in accordance with the guidelines and were approved by the National Taiwan Normal University (NTNU) Research Committee. Male C57BL/6 mice (8 weeks old, 18–22 g) were purchased from the National Laboratory Animal Center (Tainan City, Taiwan). The mice were housed in individually ventilated cages under controlled temperature (25 ± 2 °C), relative humidity (50%), and 12 h on/off light cycle with ad libitum access to food and water at the Animal House Facility of NTNU.

After one-week habituation, mice were randomly divided into 4 groups (*n* = 7). NC009-1 (40 mg/kg) or vehicle (DMSO:Cremophor EL:0.9% saline = 1:2:7) was intraperitoneally (i.p.) administrated 5 times per week for 6 weeks. Experimental parkinsonism was established by i.p. injections of 20 total doses of MPTP (25 mg/kg in 0.9% saline; Toronto Research Chemicals, Toronto, Ontario, Canada) along with probenecid (250 mg/kg in 0.1 M NaOH; Sigma-Aldrich), while control group received injections of saline. Probenecid was administered 1 h prior to MPTP administration as it decreases the clearance of MPTP and intensifies its neurotoxicity [80]. The dosage regimen was administered over 4 weeks with 5 doses per week (once daily for five consecutive days). Appropriate guidelines were abided in handling MPTP.

### 4.7. Behavioral Tests

To assess the gait performance, the CatWalkXT automatic quantitative gait analysis system (Noldus, Wageningen, The Netherlands) was utilized. The animals were placed in a walkway of 4 cm width with a glass bottom and recorded by a high-speed digital camera from below. The footprints were analyzed using the Catwalk XT 9.1 software. In addition, tail suspension test was performed for screening antidepressant-like activity in mice [24], as depression is one of the non-motor symptoms in PD [81]. This test relies on immobility as a measure of “behavioral despair” once the mouse perceives that escape from the apparatus is impossible. As described [82], each mouse was individually suspended to the edge of a table, 50 cm above the floor, by adhesive tape placed approximately 1 cm from the tip of the tail. During the test, each mouse was acoustically and visually isolated from other mice. The trials were conducted for 6 min, during which the duration of immobility was recorded. Mice were considered immobile when they hung passively and motionless.

### 4.8. HPLC Analysis of Dopamine

Levels of dopamine in striatum were determined by high performance liquid chromatography (HPLC) analysis. Briefly, the isolated brain striatum was homogenized in 500 μL of PRO-PREP^TM^ protein extraction solution (iNtRON Biotechnology Inc., Gyeonggi-do, Republic of Korea). Samples were centrifuged at 10,000 × *g* for 30 min and then filtered through a 0.45 µm syringe membrane. Dopamine from the supernatant was analyzed by the HPLC system using a C18 column with a UV detector at 254 nm. The sample was passed through the HPLC system using a mobile phase of 87.5% 90 mM of sodium phosphate, 40 mM of citric acid, 10 mM of 1-octanesulfonic acid, 3 mM of ethylenediaminetetraacetic acid (EDTA), and 12.5% acetonitrile (pH3.0) at a flow rate of 1.0 mL/min.

### 4.9. Immunohistochemistry Analysis

For the following immunohistochemistry (IHC) analysis, brains of mice were washed in PBS, fixed in 4% PFA, cryoprotected in 30% sucrose in PBS, and embedded in optimal cutting temperature compound before frozen sectioning. Three 20 μm thick sections of midbrain were cut, washed twice with PBS, and fixed in 4% PFA in PBS for 20 min at room temperature. After two rinses with PBST (PBS with 0.2% Triton) and blocking in PBST containing 3% normal serum, sections were stained with primary antibody tyrosine hydroxylase (TH, a marker for DAergic neurons) (1:50; MyBioSource #MBS421729, San Diego, CA, USA), 4-hydroxynonenal (4-HNE, an oxidative/nitrosative stress biomarker) (1:500; R&D Systems #MAB3249, Minneapolis, MN, USA), ionized calcium-binding adapter molecule 1 (IBA1, a microglial marker) (1:1000; Wako #019-19741, Osaka, Japan), or glial fibrillary acidic protein (GFAP, an astrocytic marker) (1:1000; Merck Millipore #MAB360, Burlington, MA, USA) at 4 °C overnight, followed by anti-goat (1:1000; Invitrogen #A-11055), anti-mouse (1:1000; Invitrogen #A-11001), or anti-rat (1:1000; Invitrogen #A-11006) IgG secondary antibody for 3 h at room temperature. Sections were counterstained with DAPI for 1 h. The stained cells were examined using Leica TCS SP8 confocal laser scanning microscope (Leica Microsystems, Wetzlar, Germany).

### 4.10. Neuroinflammation and Oxidative Stress Analyses in Mice

The striatum was removed immediately after the mouse was sacrificed. The tissue was homogenized by Bullet Blender (Next Advance, Averill Park, NY, USA) with zirconium oxide grinding beads (1 mm; Next Advance) for 3 min in buffer (50 mM Tris-HCl pH 8.0, 150 mM NaCl, 1 mM EDTA, 1 mM EGTA, 1% NP-40, 0.5% sodium deoxycholate, 0.1% SDS) containing protease inhibitor cocktails (Sigma-Aldrich). After 30 min incubation on ice, samples were centrifuged at 15,000× *g* for 30 min at 4 °C and supernatant collected. Proteins were quantified, separated, and blotted as described. After blocking, the membrane was probed with dopamine transporter (DAT) (1:2000; Sigma-Aldrich #MAB369), NLRP3 (1:500; Cell Signaling #15101s), CASP1 (1:500; Cell Signaling #89332s), iNOS (1:500; Cell Signaling #2982), superoxide dismutase 2 (SOD2) (1:4000; Cell Signaling #13141), NFE2-like bZIP transcription factor 2 (NRF2) (1:1000; Sigma-Aldrich #ABE413), NAD(P)H dehydrogenase, quinone 1 (NQO1) (1:2000; Sigma-Aldrich # N5288), or GAPDH (1:1000; MDBio #30000002) primary antibody at room temperature 2 h or 4 °C overnight. The immune complexes were detected as described. In addition, the levels of IL-1β, IL-6, and TNF-α in striatum were determined using mouse ELISA^TM^ kits following the manufacturer’s protocol (Invitrogen). The OD at 450 nm was detected using Multiskan Go microplate reader as described above.

### 4.11. Statistical Analysis

For each data set, three independent experiments were performed and data were expressed as the means ± standard deviation (SD). Differences between groups in the cell experiments were evaluated by one-way analysis of variance (ANOVA) with a post hoc Bonferroni test. In the animal experiments, the Kruskal–Wallis test (nonparametric test to compare unmatched groups) with a post hoc Dunn’s test was applied to compare the differences (*n* = 5–7). Nonparametric Spearman’s correlations were applied to evaluate the relationship between the 4-HNE/TH ratio in immunohistochemistry and parameters of behavioral tests. All *p* values were two-tailed, with values lower than 0.05 considered being statistically significant.

## 5. Conclusions

In the present study, we show that NC009-1 exerts neuroprotective effects by modulating inflammatory and anti-oxidative pathways (Figure 6). It is of note that NC009-1 demonstrates good bioavailability and BBB penetration potential [13], further enhancing its potential for translation to clinical practice. Future studies in other animal models of PD will be necessary to validate its potential for treating PD before moving toward to clinical trials.

## Figures and Tables

**Figure 1 ijms-24-02642-f001:**
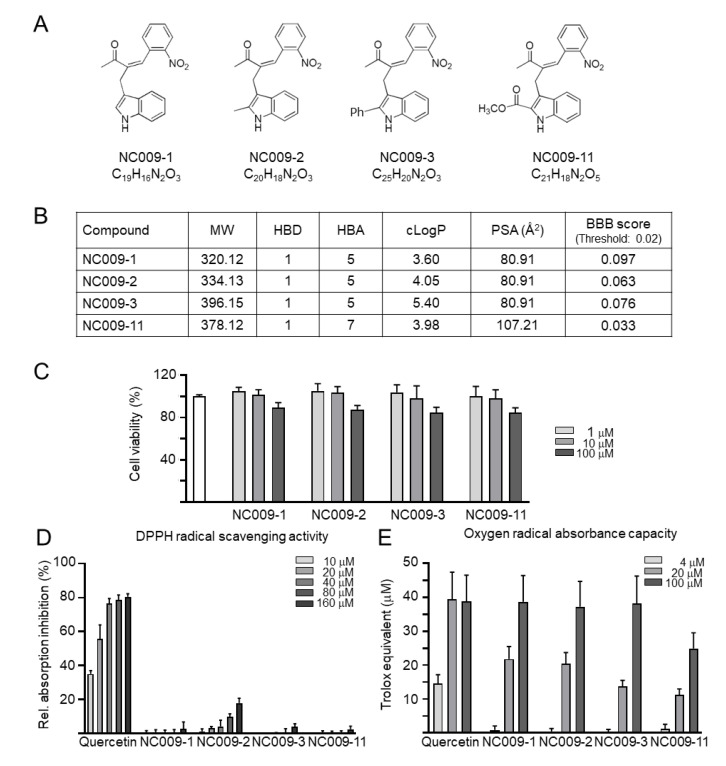
NC009 compounds. (**A**) Structure and formula of NC009-1, -2, -3, and -11. (**B**) Molecular weight (MW), hydrogen bond donor (HBD), hydrogen bond acceptor (HBA), calculated octanol–water partition coefficient (cLogP), polar surface area (PSA), and blood–brain barrier (BBB) permeation score of NC009 compounds. (**C**) Cytotoxicity of NC009 compounds against human HMC3 cells examined by MTT assay. Cells were treated with test compound (1–100 μM) and cell viability was measured the next day (*n* = 3). To normalize, the relative viability of untreated cells was set at 100%. (**D**) DPPH free radical scavenging activity on DPPH (10–160 μM) and (**E**) oxygen radical absorbance capacity (4–100 μM) of quercetin (as a positive control) and NC009 compounds (*n* = 3).

**Figure 2 ijms-24-02642-f002:**
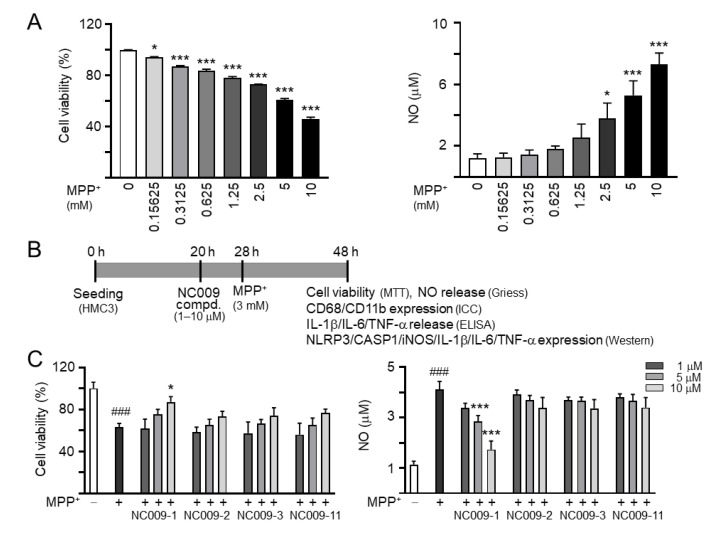
MPP^+^-induced inflammation of HMC3 microglia and anti-inflammatory potentials of NC009 compounds. (**A**) Cytotoxicity of MPP^+^ against HMC3 cells by MTT and NO release assays. Cells were treated with MPP^+^ (0–10 mM) and cell viability and NO release in culture medium was measured the following day (*n* = 3). To normalize, the relative cell viability in MPP^+^-untreated (inactive) cells was set at 100%. *p* values: comparisons between inactive and activated cells (* *p* < 0.05, *** *p* < 0.001). (**B**) Experimental flow chart. HMC3 cells were plated on day 1. After 20 h, cells were pre-treated with test compounds (1–10 μM) for 8 h, followed by MPP^+^ (3 mM) treatment for 20 h. On day 3, the HMC3 cells were examined for cell viability (MTT assay) and NO release (Griess assay) in culture medium. In addition, cells with 10 μM compound treatment were examined for CD68 and CD11B expression (ICC staining), IL-1β, IL-6, TNF-α release (ELISA), and NLRP3, CASP1, iNOS, IL-1β, IL-6, and TNF-α expression (Western). (**C**) Cell viability and NO release assays (*n* = 3). The relative cell viability in MPP^+^-untreated cells was normalized as 100%. (**D**) CD68 and CD11B expression (*n* = 3). Cells were analyzed by immunofluorescence using antibodies against CD68 (yellow) and CD11B (green). Cell nuclei were counterstained with DAPI (blue). (**E**) IL-1β, IL-6, and TNF-α release assays (*n* = 3). (**F**) NLRP3, CASP1, iNOS, CREB, IL-1β, IL-6, and TNF-α levels analyzed by immunoblotting using GAPDH as a loading control (*n* = 3). To normalize, protein expression level in MPP^+^-untreated cells was set at 100%. (**C**–**F**), *p* values: comparisons between MPP^+^-untreated and treated cells (^#^ *p* < 0.05, ^##^ *p* < 0.01, ^###^ *p* < 0.001), or with and without compound addition (* *p* < 0.05, ** *p* < 0.01, *** *p* < 0.001).

**Figure 3 ijms-24-02642-f003:**
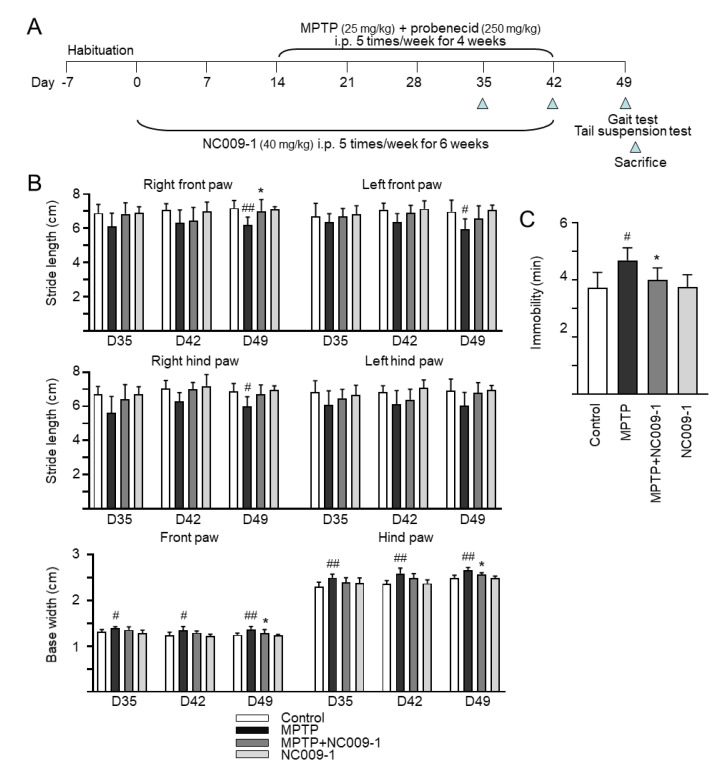
Neuroprotective effects of NC009-1 in MPTP-induced mouse model of PD. (**A**) Experimental protocol. After one week’s habituation, vehicle or NC009-1 (40 mg/kg) was i.p. administrated into control and MPTP groups or NC009-1 and MPTP+NC009-1 groups (*n* = 7) 5 times per week for 6 weeks. Two weeks after vehicle/NC009-1 administration, MPTP and MPTP+NC009-1 groups received i.p. injections of MPTP (25 mg/kg) along with probenecid (250 mg/kg) for 4 weeks (5 doses/week), while control and NC009-1 groups received injections of saline. Mice were subjected to gait test on days 35, 42, and 49, and tail suspension test on day 49. Next day, mice were sacrificed for HPLC, IHC (Figure 4), and Western (Figure 5) analyses. (**B**) Gait test to assess gait variability. Stride length (right front paw, right hind paw, left front paw, and left hind paw) and base width (front paw and hind paw) were measured as the distance between two paw prints. (**C**) Tail suspension test to assess immobility. Duration of immobility within 6 min was recorded. (**B**,**C**), *p* values: comparisons between MPTP and control (^#^ *p* < 0.05, ^##^ *p* < 0.01), or NC009-1-treated and untreated MPTP mice (* *p* < 0.05).

**Figure 4 ijms-24-02642-f004:**
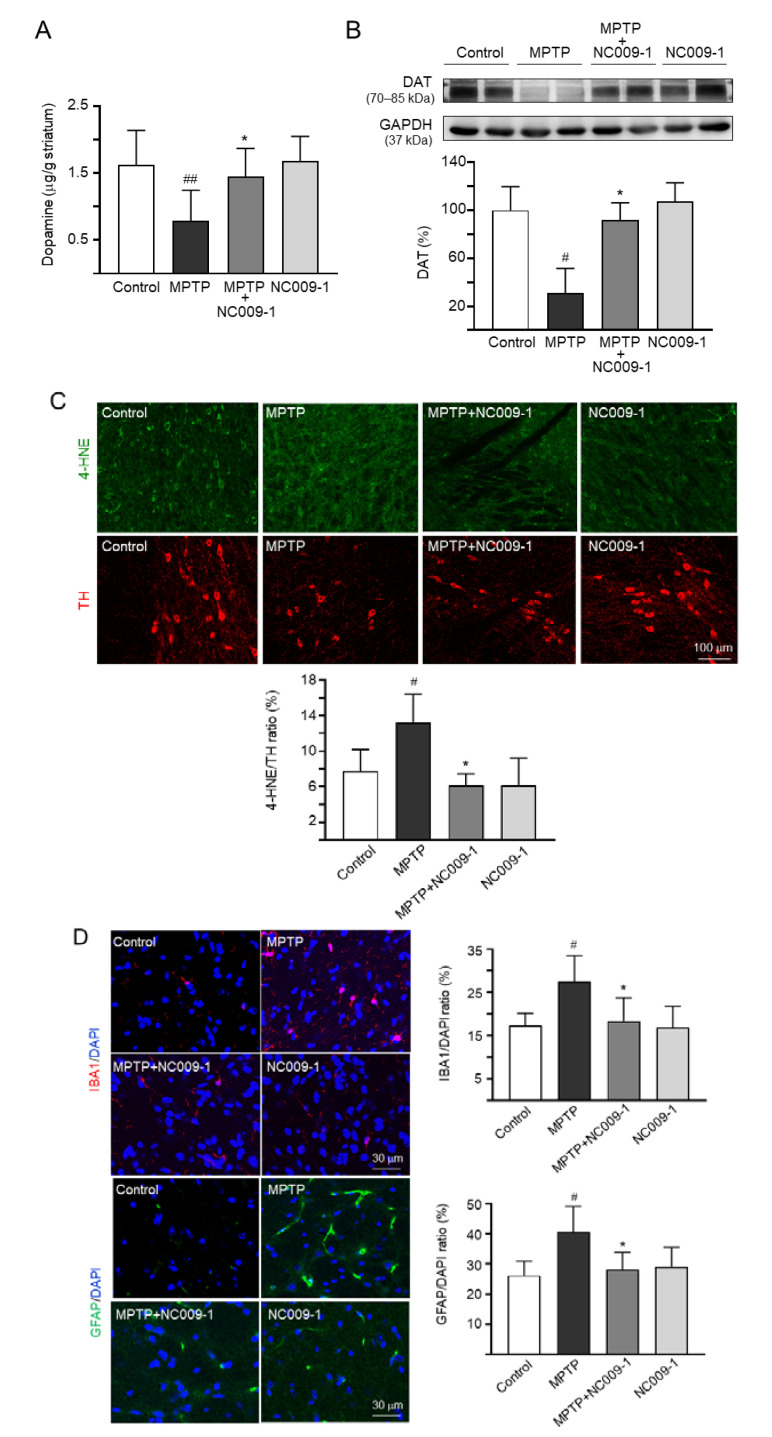
NC009-1 attenuated the loss of striatal dopamine, oxidative damage, and neuroinflammation in MPTP-PD mice. (**A**) HPLC quantitation of striatal dopamine (*n* = 7). (**B**) Immunoblotting of striatal dopamine transporter (DAT) (*n* = 6). (**C**) IHC stains of TH (red) and 4-HNE (green) (*n* = 5). Quantitation of TH^+^ neurons with oxidative damage, based on TH and 4-HNE co-localization, is shown below. (**D**) IHC stains of IBA1 (red) and GFAP (green) at ventral midbrain (*n* = 5). Nuclei were counter-stained with DAPI (blue). Shown left of the images are quantitation of IBA1 and GFAP labeling. *p* values: comparisons between MPTP and control (^#^ *p* < 0.05, ^##^ *p* < 0.01), or NC009-1-treated and untreated MPTP mice (* *p* < 0.05).

**Figure 5 ijms-24-02642-f005:**
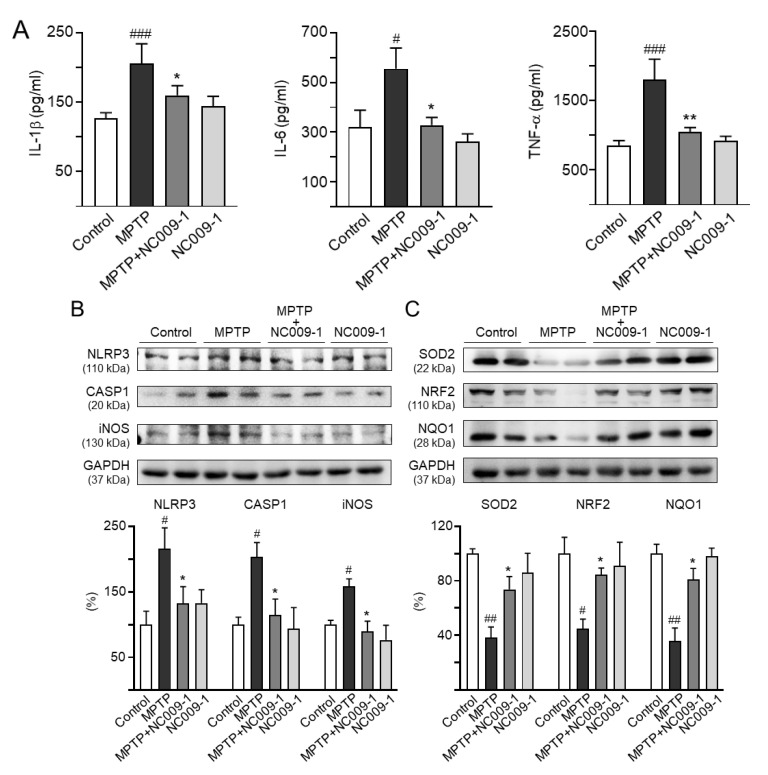
NC009-1 down-regulated neuroinflammation and up-regulated cellular redox signaling in MPTP-treated mice. (**A**) Expression levels of striatal IL-1β, IL-6, and TNF-α analyzed by ELISA (*n* = 6). Expression levels of (**B**) NLRP3, CASP1, iNOS, and (**C**) SOD2, NRF2, and NQO1 in striatum analyzed by Western blot (*n* = 6). GAPDH was included as a loading control. To normalize, the relative NLRP3, CASP1, iNOS, SOD2, NRF2, and NQO1 of control mice was set as 100%. *p* values: comparisons between MPTP and control (^#^ *p* < 0.05, ^##^ *p* < 0.01, ^###^ *p* < 0.001), or NC009-1-treated and untreated MPTP mice (* *p* < 0.05, ** *p* < 0.01).

**Figure 6 ijms-24-02642-f006:**
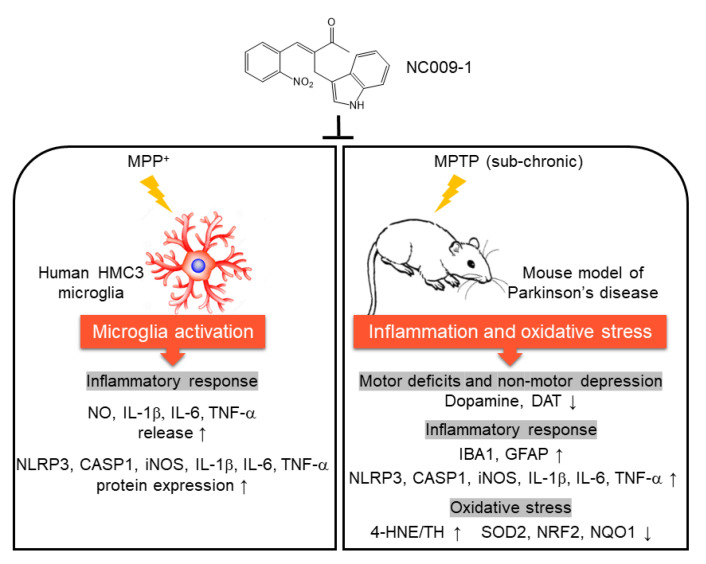
Graphical summary.

## Data Availability

All data generated or analyzed during the current study are available from the corresponding author on reasonable request.

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
