# Peer review of "Investigating Therapeutic Effects of Indole Derivatives Targeting Inflammation and Oxidative Stress in Neurotoxin-Induced Cell and Mouse Models of Parkinson’s Disease"

_ijms, 2023, doi:10.3390/ijms24032642_

Round 1

Reviewer 1 Report

Basic Reporting and Comments:

Authors have identified the effect of indole derivative NC009-1 in protecting against neuroinflammation and oxidative stress responsible for causing Parkinson's disease. They have done a detailed mechanistic study by analyzing the various neurodegeneration markers. Overall, the study has been done thoroughly with a good experimental strategy. 

1. What is the major difference between the indole derivative NC001-8 and NC001-9? Is it structurally different which can affect its chemical activity? As the authors have already published that NC008-1 also protects against neurotoxicity earlier. What was the main reason for not combining these two derivatives in one single study as a comparative analysis?

2. It would be good to include the synthesis process of these indole derivatives in the supplementary material.

3. Line 115, it would be more appropriate to use the word "significantly" instead of effectively as it is a statistically significant increase in cell viability.

4. Fig 2D, is it also 50 um for the DIC image as well?

5. It is advised to include the full labeling NC009-1, NC009-2, NC009-3, and NC009-11 in Fig 2 and 3, just like Figure 1 for better readability.

6. Even 5 uM is showing a significant decrease in NO production in Fig 2B. It can also be mentioned in the text as the effect of lower concentration is always considered better.

7. Fig 3B, can be subdivided based on different paws, as it is appearing very crowded to see clear differences.

8. Western blot quality can be improved for Fig 5A.

9. It would be good to include the software usage details for ChemDraw and BBB predictor pertaining to your specific study in the methods section or in the supplementary material.

10. For the graphical summary, it is advised to turn the figure upside down, NC009-1 should be at the top and then the mechanism can be down to it for a better view. Minor suggestion.

Author Response

  1. What is the major difference between the indole derivative NC001-8 and NC009-1? Is it structurally different which can affect its chemical activity? As the authors have already published that NC001-8 also protects against neurotoxicity earlier. What was the main reason for not combining these two derivatives in one single study as a comparative analysis?

Response: The compounds NC009-1, NC009-2, NC009-3 and NC009-11 were synthesized through iodine-catalyzed C-alkylation of indoles, as reported by Ramesh et al. in 2009 [78]. However, the synthesis and structure of NC001-8, as reported by Janreddy et al. in 2011, were different and not derived from NC009-1. Therefore, we focused our evaluation on the anti-inflammatory and neuroprotective properties of NC009-1 and its derivatives only.

Janreddy D, Kavala V, Bosco JWJ, Kuo CW, Yao CF. An easy access to carbazolones and 2,3-disubstituted indoles. Eur J Org Chem. 2011; 12(12):2360–2365.

  1. It would be good to include the synthesis process of these indole derivatives in the supplementary material.

Response: We briefly mentioned the synthesis process of these indole derivatives in Materials and Methods (lines 337-338): Indole derivatives NC009-1, -2, -3, -11 were synthesized by iodine-catalyzed C-alkylation of indoles as previously described [78].

  1. Line 115, it would be more appropriate to use the word "significantly" instead of effectively as it is a statistically significant increase in cell viability.

Response: We made the change in line 120 as suggested.

  1. Fig 2D, is it also 50 um for the DIC image as well?

Response: Yes. We labeled DIC image as well (line 139).

  1. It is advised to include the full labeling NC009-1, NC009-2, NC009-3, and NC009-11 in Fig 2 and 3, just like Figure 1 for better readability.

Response: We included the full labeling of test compounds (lines 138, 140 and 141) as suggested.

  1. Even 5 µM is showing a significant decrease in NO production in Fig 2B. It can also be mentioned in the text as the effect of lower concentration is always considered better.

Response: We revised the sentence as suggested (lines 121-122): In addition, NC009-1 at 5−10 µM decreased release of NO in cell culture medium (from 4.1 µM to 2.8−1.7 µM, p < 0.001).

  1. Fig 3B, can be subdivided based on different paws, as it is appearing very crowded to see clear differences.

Response: We revised Fig. 3B (lines 183-184) as suggested.

  1. Western blot quality can be improved for Fig 5A.

Response: We improved Western blot quality for NLRP3, CASP1 and iNOS (line 254).

  1. It would be good to include the software usage details for ChemDraw and BBB predictor pertaining to your specific study in the methods section or in the supplementary material.

Response: We included software usage details as suggested: ChemDraw Ultra 12.0 (line 350), online BBB predictor version 0.90 (line 352).

  1. For the graphical summary, it is advised to turn the figure upside down, NC009-1 should be at the top and then the mechanism can be down to it for a better view.

Response: We revised graphical summary as suggested (line 521).

Reviewer 2 Report

The authors of this research paper tested the neuroprotective effects of indole derivatives NC009-1, -2, -3, -11 in a chemical model of Parkinson's disease, specifically the MPTP model. This is a follow up study of previous research studies carried out by the authors showing the feasibility of indole derivatives to reduce inflammation and microglia activation in cell culture models.

Overall, I found this study to be of high interest to neurodegeneration researchers and of high relevance for developing new therapeutics to treat Parkinson's disease. The figures are of high quality and all the image-based quantifications and controls are provided and rigorous.

However, I have some modest and minor concerns if the authors can address them to make the paper publishable as listed below:

1. Please provide more background regarding the synthesis and rationale for the use of indole derivatives. Can you provide more detail background on the parent, unmodified molecule? What was the rationale for making the side chain modifications and more on the click chemistry approach used in the methods.

2. As an extension of point #1, it seems that compound  -11 has no neuroprotective properties or obvious antioxidant effects. What was the rationale for using this compound in this study? Was it used as a negative control (in the case the side chains modifications block its antioxidant activities)? More explanation is needed.

3. Regarding the figures: Figure 4D, the MPTP+ treatment does not seem to show an increase in Iba1 expression (microglia) and the co-treatment with the first compound (-1) does not seem to reduce the mean fluorescence. Can the authors provide better panels that are representative of the quantification?

4. None of the figure legends have sufficient details regarding the statistical analysis done. The authors should indicate the number of experiments and replicates, animals used and statistics (post-hoc analyses) and whether bonferroni correction was used used per figure legend.

5. The Methods section indicate that Tukey's posthoc tests were used following ANOVA for all the data shown in the paper. As a follow up of point #4, parametric tests for post hoc analyses cannot be used for animal data since the distribution is usually non-parametric. Mann-Whitney or Wilcoxon statistical analyses should be used.

6.  The authors show data on motor coordination of MPTP-treated animals and co-treated with the indole compound -1. Gait analyses is appropriate as one test of motor coordination but can be complemented with other studies. What about rotarod and beam balance tests? These tests are more sensitive and provide a more sensitive scale for motor coordination than the gait analysis shown. Much of the motor data shown in figure 3 is highly variable with very modest results. A more sensitive motor test will be preferable or the authors should provides strong rationale for only doing one motor test.

7. Figure 4 shows some data with TH neurons in MPTP-treated mice. However,  did the authors do any stereology of TH neurons to associate motor symptoms (or recovery from motor symptoms) in the mice? This is very important to determine whether intraperitoneal administration of indole derivatives can reverse neurodegeneration, and to know how much neurodegeneration is achieved with chronic MPTP dosing is.

8. The authors should provide more details on how the image-based data was quantified (mean intensity, integrated density) and was the data normalized to cell number? The data is normalized to DAPI ratio but more details are needed to understand this ratio. This is not clear.

9. Have the authors studied the brain and serum bioavailability of indole compound -1 to see how much of it can reach the brain after intraperitoneal administration? If this has been done in the past, then cite the studies.

10. Finally, the authors should provide a figure model to show the mechanism of action of indole derivatives in deducing inflammation, microglia activation and motor symptoms.

Author Response

  1. Please provide more background regarding the synthesis and rationale for the use of indole derivatives. Can you provide more detail background on the parent, unmodified molecule? What was the rationale for making the side chain modifications and more on the click chemistry approach used in the methods.

Response: We briefly mentioned the synthesis process of these indole derivatives in Materials and Methods (lines 337-338): Indole derivatives NC009-1, -2, -3, -11 were synthesized by iodine-catalyzed C-alkylation of indoles as previously described [78]. We also added sentences in Introduction to provide rationale for the use of indole derivatives (lines 60-63 & 72-75): Indole is an aromatic heterocyclic compound with a wide range of biological activities. Its chemical reactivity makes it a suitable candidate for modification, leading to the development of various novel derivatives with potential as drug candidates for the treatment of various diseases [11]. ….. In the present study, we aimed to investigate the neuroprotective potential of NC009-1 and derivative compounds with methyl, phenyl or methyl formate substituent present on the benzene ring in MPP+-activated human microglial HMC3 cells and/or sub-chronic MPTP-induced mouse model of PD.

[11] Kumar, D.; Sharma, S.; Kalra, S.; Singh, G.; Monga, V.; Kumar, B. Medicinal perspective of indole derivatives: Recent developments and structure-activity relationship studies. Curr. Drug Targets. 2020, 21, 864–891.

  1. As an extension of point #1, it seems that compound-11 has no neuroprotective properties or obvious antioxidant effects. What was the rationale for using this compound in this study? Was it used as a negative control (in the case the side chains modifications block its antioxidant activities)? More explanation is needed.

Response: NC009-11 compounds at 20−100 μM concentration displayed 11−25 μM trolox equivalent antioxidant capacity based on the trapping of peroxyl radicals. Although this antioxidant capacity was lower compared to other compounds, we included all four compounds in the HMC3 cell study for comparison (lines 96-97): For comparison, all four NC009 compounds were included in the following HMC3 cell study.

  1. Regarding the figures: Figure 4D, the MPTP+ treatment does not seem to show an increase in Iba1 expression (microglia) and the co-treatment with the first compound (-1) does not seem to reduce the mean fluorescence. Can the authors provide better panels that are representative of the quantification?

Response: We provided better panels as suggested (line 224).

  1. None of the figure legends have sufficient details regarding the statistical analysis done. The authors should indicate the number of experiments and replicates, animals used and statistics (post-hoc analyses) and whether bonferroni correction was used used per figure legend.

Response: We revised 4.11. Statistical Analysis (lines 506-512) to address questions regarding statistical analysis: Differences between groups in the cell experiments were evaluated by one-way analysis of variance (ANOVA) with a post hoc Bonferroni test. In the animal experiments, Kruskal–Wallis test (nonparametric test to compare unmatched groups) with a post hoc Dunn’s test was applied to compare the differences (n = 5–7). Nonparametric Spearman’s correlations were applied to evaluate the relationship between the 4-HNE/TH ratio in immunohistochemistry and parameters of behavioral tests. We also added sufficient details regarding the number of experiments and animals used in figure legends 2-5.

  1. The Methods section indicate that Tukey's posthoc tests were used following ANOVA for all the data shown in the paper. As a follow up of point #4, parametric tests for post hoc analyses cannot be used for animal data since the distribution is usually non-parametric. Mann-Whitney or Wilcoxon statistical analyses should be used.

Response: We revised 4.11. Statistical Analysis (lines 506-512) as stated.

  1. The authors show data on motor coordination of MPTP-treated animals and co-treated with the indole compound -1. Gait analyses is appropriate as one test of motor coordination but can be complemented with other studies. What about rotarod and beam balance tests? These tests are more sensitive and provide a more sensitive scale for motor coordination than the gait analysis shown. Much of the motor data shown in figure 3 is highly variable with very modest results. A more sensitive motor test will be preferable or the authors should provide strong rationale for only doing one motor test.

Response: We had assessed the effect of NC009-1 on motor coordination using the pole climbing and rotarod tests. However, both tests did not display motor deficits in MPTP-treated mice. As described in literature (Wang et al., 2019), sub-chronic MPTP treatment may not induce abnormal findings in pole climbing and rotarod tests, even though MPTP does trigger oxidative stress and inflammatory reactions.

Wang, L.Y.; Yu, X.; Li, X.X.; Zhao, Y.N.; Wang, C.Y.; Wang, Z.Y.; He, Z.Y. Catalpol exerts a neuroprotective effect in the MPTP mouse model of Parkinson's disease. Front. Aging Neurosci. 2019, 11, 316.

  1. Figure 4 shows some data with TH neurons in MPTP-treated mice. However, did the authors do any stereology of TH neurons to associate motor symptoms (or recovery from motor symptoms) in the mice? This is very important to determine whether intraperitoneal administration of indole derivatives can reverse neurodegeneration, and to know how much neurodegeneration is achieved with chronic MPTP dosing is.

Response: We added a sentence to address this (lines 211-213): 4-HNE/TH ratio demonstrated correlation with base width of front paw (day 49: r = 0.548, p = 0.012), stride length of right front paw (day 49: r = -0.492, p = 0.028), or time of immobility (r = 0.468, p = 0.038).

  1. The authors should provide more details on how the image-based data was quantified (mean intensity, integrated density) and was the data normalized to cell number? The data is normalized to DAPI ratio but more details are needed to understand this ratio. This is not clear.

Response: We provided details as requested (lines 403-406): All fluorescence image intensities were normalized to the number of DAPI-positive signal cells to compute the mean cell integrated intensity. Three independent experiments were performed in triplicate, with around 7500 cells analyzed from each sample.

  1. Have the authors studied the brain and serum bioavailability of indole compound -1 to see how much of it can reach the brain after intraperitoneal administration? If this has been done in the past, then cite the studies.

Response: Thank you for raising this important question. We did not examine the brain and serum bioavailability of NC009-1 after intraperitoneal administration before due to resource constraints.

  1. Finally, the authors should provide a figure model to show the mechanism of action of indole derivatives in deducing inflammation, microglia activation and motor symptoms.

Response: We modified Graphical summary (Figure 6, line 521) as suggested.

Round 2

Reviewer 2 Report

The authors addressed my concerns and have strengthened the quality of the presentation of the data as well its interpretation. Congratulations!